# GSK2801 Reverses Paclitaxel Resistance in Anaplastic Thyroid Cancer Cell Lines through MYCN Downregulation

**DOI:** 10.3390/ijms24065993

**Published:** 2023-03-22

**Authors:** Elisabetta Molteni, Federica Baldan, Giuseppe Damante, Lorenzo Allegri

**Affiliations:** 1Department of Medicine, University of Udine, Via Chiusaforte, 33100 Udine, Italy; 2Institute of Medical Genetics, Academic Hospital of Udine, Azienda Sanitaria Universitaria Integrata di Udine, 33100 Udine, Italy

**Keywords:** thyroid cancer, paclitaxel, drug resistance, bromodomain, BAZ2A, BAZ2B, BRD9, GSK2801, MYCN, VPC-70619

## Abstract

Anaplastic thyroid cancer (ATC) is a very rare, but extremely aggressive form of thyroid malignancy, responsible for the highest mortality rate registered for thyroid cancer. Treatment with taxanes (such as paclitaxel) is an important approach in counteracting ATC or slowing its progression in tumors without known genetic aberrations or those which are unresponsive to other treatments. Unfortunately, resistance often develops and, for this reason, new therapies that overcome taxane resistance are needed. In this study, effects of inhibition of several bromodomain proteins in paclitaxel-resistant ATC cell lines were investigated. GSK2801, a specific inhibitor of BAZ2A, BAZ2B and BRD9, was effective in resensitizing cells to paclitaxel. In fact, when used in combination with paclitaxel, it was able to reduce cell viability, block the ability to form colonies in an anchor-independent manner, and strongly decrease cell motility. After RNA-seq following treatment with GSK2801, we focused our attention on MYCN. Based on the hypothesis that MYCN was a major downstream player in the biological effects of GSK2801, we tested a specific inhibitor, VPC-70619, which showed effective biological effects when used in association with paclitaxel. This suggests that the functional deficiency of MYCN determines a partial resensitization of the cells examined and, ultimately, that a substantial part of the effect of GSK2801 results from inhibition of MYCN expression.

## 1. Introduction

Thyroid cancer (TC) is the most common endocrine carcinoma, accounting for 1–2% of cancer cases worldwide [1]. Thyroid cancer originates from follicular epithelial cells or parafollicular C cells. The ones derived from the follicular cells are classified into four histological types: papillary thyroid cancer (PTC, 80–85%), follicular thyroid cancer (FTC, 10–15%), poorly-differentiated thyroid cancer (PDTC, <2%), and anaplastic thyroid cancer (ATC, <2%) [2]. PTC and FTC account for more than 95% of tumors and are mostly responsive to current treatment based on surgery and subsequent radioiodine therapy, with an overall survival rate of more than 90% within 10 years. The situation is different regarding ATC, which is a very rare but extremely aggressive form of thyroid neoplasm. In fact, although it accounts for a very small percentage of all thyroid malignancies (less than 2%), it is mostly diagnosed as stage IV disease, is often unresectable, and fails to respond to radioactive iodine ablation because of its undifferentiated phenotype due to loss of thyroid-specific gene expression. Therefore, ATC is responsible for 20–50% of thyroid cancer mortality [3]. Stage IVA and IVB (when resectable) ATCs are treated by surgical resection followed by intensity-modulated radiation therapy (IMRT) and taxane-based chemotherapy alone or in combination with platinum or anthracycline [4]. For not-resectable stage IVB ATC, two classes of FDA-approved drugs are presently available: tyrosine kinase inhibitors (TKIs) and mutation-specific drugs [5]. However, in some patients, both of these strategies fail [6]. In this case, the current treatment is still palliative surgery and systemic chemotherapy, the latter consisting of monotherapy with taxanes or in combination with carboplatin or anthracyclines [7]. Taxanes, therefore, still represent an important player in counteracting ATC or slowing its progression. Unfortunately, treatment with these compounds easily induces drug resistance. Thus, in order to improve the prognosis of this very aggressive neoplasia, it is essential to understand the biological and/or pharmacological mechanisms to overcome the resistance toward taxanes. A well-established approach in counteracting cancer cell resistance to treatments is the use of combination therapies acting on different pathways synergistically or that are able to resensitize cancer cells to monotherapy [8]. More specifically, regarding the use of taxanes, synergistic antitumor effects have already been demonstrated in the case of multiple treatment [9,10].

Bromodomains (BRDs) are evolutionarily conserved protein–protein interaction modules with different catalytic and scaffolding functions in a wide range of proteins and tissues. The bromodomain-containing family proteins represent an important class of histone modification readers that recognize acetylated lysine residues [11,12]. This class of protein is often dysregulated in cancer, and many cancer-causing mutations have been mapped to the BRDs of these same proteins [13]. The relevance of BRD proteins in thyroid cancer is known and is well described in the literature. Their blocking by specific inhibitors such as JQ1 and AZD5153 has been shown to suppress tumor growth in vitro and in vivo [14,15,16]. The human proteome encodes 61 BRD modules found in 42 different proteins [12], which can be classified into eight subfamilies based on their sequence and structure [17]. BRD9, a member of subfamily IV, is a protein identified as a dedicated member of the mammalian SWI/SNF complex; it maintains and facilitates oncogenic transcription directly by recognizing the acetylated lysine on post-translationally modified histone proteins, contributing to cancer cell proliferation and survival [18]. Recently, a research group identified BRD9 as an HR (homologous recombination) regulator that facilitates the functions of RAD54 and RAD51 by acting as a bridge between the two proteins [13]. BRDs are also found in the family of BRD proteins adjacent to the zinc finger (BAZ) domain, which in humans are encoded by four related genes: *BAZ1A, BAZ1B, BAZ2A*, and *BAZ2B* [19]. BAZ2A and BAZ2B are members of the bromodomain family V [20]. BAZ2A (also known as TIP5) is a key subunit of the nucleolar remodelling complex (NoRC) that has a role in RNA-dependent noncoding silencing [19]. Recent work has implicated BAZ2A in prostate cancer (PCa) [17]. BAZ2A is highly expressed in metastatic tumors compared with primary and localized tumors; it is involved in cell proliferation, viability, and invasion, and it represses genes frequently silenced in aggressive PCa [21]. BAZ2A also induces aberrant gene silencing in PCa in association with enhancer of zest homologue 2 (EZH2), whose altered expression is correlated with malignancy and poor prognosis [22,23]. The bromodomain of BAZ2B shares a similar Kac binding site (76% sequence identity and 93% similarity) with its paralog BAZ2A. The physiological function of BAZ2B is still unclear [19]. However, high expression levels of BAZ2B have been associated with poor outcome in pediatric B-cell acute lymphoblastic leukemia (B-ALL) [23]. In the last few years, small molecule bromodomain inhibitors (BDIs) have increased in usability for their effective anticancer activity and have emerged as a promising class of anticancer drugs [24]. Chen et al. developed a novel inhibitor of BDs from a nonselective micromolar BAZ/BET inhibitor. This inhibitor, GSK2801, is a potent and selective ligand for BAZ2A and BAZ2B bromodomains, suitable for use as a chemical probe for small molecules. GSK2801 shows high selectivity for BAZ2A and BAZ2B, and also binds with lower affinity to BRD9 [25]. The antiproliferative effect of this inhibitor in synergy with BET inhibitors has already been tested in a triple negative breast cancer cell line [24,26]. Furthermore, through using the ChIP-seq approach, it was recently discovered that GSK2801 significantly disrupts chromatin binding of flag-tagged BAZ2A or BAZ2B in mouse liver H2.35 cells [27].

## 2. Results

Since in a previous study, mRNA-seq analysis had shown expression of GSK2801 targets (*BAZ2A, BAZ2B* and *BRD9*) in SW1736 and 8505C paclitaxel-resistant cells [3], we wondered whether these bromodomain protein actions represent a mechanism of resistance to taxane treatment in this model. Figure 1 shows that in paclitaxel-resistant ATC cells (8505C-PTX and SW1736-PTX), the expression level of *BRD9* is increased compared with non-resistant parental lines both at the mRNA and protein levels (Figure 1, panel A and C). In contrast, gene expression analysis and Western blot analysis of *BAZ2A* and *BAZ2B* show no significant change in resistant cells.

### 2.1. Effects of GSK2801 on ATC Cell Viability, Apoptosis, Colony Formation, and Cell Migration

In this study, the biological effects of the BAZ2A, BAZ2B, and BRD9 specific inhibitor GSK2801 on two resistant ATC cell lines were investigated. In a first experimental setting, we evaluated the effect on cell viability of different doses of GSK2801 in two paclitaxel-sensitive ATC cell lines (SW1736 and 8505C, Figure 2 panel A) and in two paclitaxel-resistant ATC cell lines (SW1736-PTX and 8505C-PTX to GSK2801, Figure 2, panel B), alone or in combination with paclitaxel (Figure 2, panel C). Treatments with GSK2801 show very modest effects in paclitaxel-sensitive cells or in SW1736-PTX and 8505C-PTX cell lines; moreover, there is no dose-response effect as even when increasing the dose up to 40 μM, no effect is shown in SW1736, 8505C, and SW1736-PTX cells, or a mild effect in 8505C-PTX cells is observed. Since these are resistant cells, treatment with low concentrations of paclitaxel obviously has no effect. However, as shown in Figure 2 panel C, combinations of paclitaxel and GSK2801 at different concentrations result in a strong decrease in cell viability, which suggests a partial resensitization of the cells to paclitaxel. The combination used for the following experiments is the one that showed the greatest reduction in viability (PTX 200 nM and GSK2801 20 μM). Given the reduction in cell viability after combined treatment with paclitaxel and GSK2801, we evaluated whether it could be due to apoptotic phenomena. For this purpose, cleaved PARP levels were assessed as a marker of apoptosis. Figure 2, panels D and E, shows the effects of simultaneous administration of paclitaxel and GSK2801 on cleaved PARP protein levels. As can be seen from the figure, a significant increase in apoptosis was observed in both cell lines following treatment.

After determining the treatment dose of GSK2801 alone and in combination with paclitaxel causing an important effect on cell viability, we investigated the effect of this molecule on the ability to form colonies in an anchorage-dependent way in the two cell lines. The two cell lines react differently when treated with GSK2801 alone. In 8505C-PTX cells, as shown in Figure 3 (panels A and C), the number of colonies formed after treatment with GSK2801 alone was greatly reduced compared with paclitaxel alone. When administered together with paclitaxel, GSK2801 had a greater effect in reducing colonies, which was significant compared with both single treatment and control. Regarding SW1736-PTX cells (Figure 3 (panels B and D)), a significant reduction was observed only in combination treatment, whereas when treated with GSK2801 alone, the number of colonies was very similar to that obtained in the paclitaxel treatment situation.

### 2.2. Effects of GSK2801 on In Vitro Tumor Cell Aggressiveness

To investigate in more detail the effect of this drug on the aggressive behavior of cancer cells, we examined the ability of cells to form colonies in an anchorage-independent way. A well- established in vitro method to characterize this competence is soft agar assay [28]. As shown in Figure 4 (panels A and B), after treatment with 20 μM GSK2801 alone, we observed a strong significant reduction in the number of colonies only in treated SW1736-PTX cells (Figure 4 panel B), while in combined treatment (200 nM PTX and 20 μM GSK2801), the number of colonies was significantly reduced in both 8505C-PTX and SW1736-PTXcells.

Another capacity related to tumor aggressiveness is the migration ability. We performed a scratch assay in order to explore this capacity of paclitaxel-resistant ATC cells in the presence or absence of treatment. The 2D wound healing assay is essentially based on the disruption of the cell monolayer at confluence, which generates a cell-free region, which can be re-covered by cells based on their ability to migrate [29]. Figure 4, panels C–E, shows the analysis of the data obtained through this experiment in the time frame 0–6 (hours). In both cell lines, treatment with GSK2801 leads to acceleration of wound closure. In contrast, when cells were treated with both paclitaxel and GSK2801, their closure rate was strongly reduced compared with paclitaxel-only treatment.

### 2.3. Effects of GSK2801 on Gene Expression in Paclitaxel-Resistant ATC Cells

To better understand the molecular mechanisms behind the biological effects exhibited by GSK2801 on paclitaxel-resistant ATC cells, a high-throughput RNA-sequencing analysis after GSK2801 treatment was performed. To detect direct effects of BAZ1B, BAZ2B, and BRD9 inhibition, GSK2801 or vehicle (DMSO) was administered at a concentration of 20 μM for an early time of 6 h. RNA extracted from four biological replicates for each experimental point was pulled and then subjected to high-throughput RNA-sequencing analysis, and for each cell line, samples treated with GSK2801 were compared against those that had received only DMSO. In SW1736-PTX cells, treatment with GSK2801 resulted in upregulation of 11495 line-specific genes and downregulation of 2518 line-specific genes. In 8505C-PTX cells, on the other hand, the same treatment resulted in increased expression of 2815 genes, while inducing reduced expression in 321 genes line-specific in both cases. Comparing commonly upregulated genes and commonly downregulated genes in the two cell lines, the administration of GSK2801 for 6 h increased the expression of 250 genes and reduced expression of 155 genes (Figure 5, panel A). Overall, the effects of early treatment that inhibits BAZ2A, BAZ2B, and BRD9 appear to have an activating effect with an increase in gene expression in both lines, most evident in SW1736-PTX, probably because these cells when untreated (DMSO) showed lower gene expression levels (Figure 5, panel B). Of particular interest was the analysis of commonly up- and downregulated genes in the two different cell lines. As shown in Figure 5 (panel C), the two pathways most affected by GSK2801 treatment were the cadherin signaling and the WNT signaling ones. Some genes that were part of these two pathways were upregulated in both cell lines, while others were downregulated. More specifically, the individual genes showing a change in expression in each pathway, or common to the two, are listed in Table 1.

Assuming greater cause–effect linearity between GSK2801 treatment and reduced expression levels rather than with increased gene expression, we focused on genes commonly downregulated in the two cell lines. After assessing the levels of gene expression reduction (Log2), their cellular function, basal expression levels, and grade of relevance in involvement in tumor, we chose *MYCN* as an important gene whose expression is reduced in both cell lines.

### 2.4. ChIP Analysis of the Effects of GSK2801 on the Binding of BRD9 to Its Targets

In order to evaluate the effects of GSK2801 on the interaction of BRD9 with its targets, a ChIP analysis was conducted using an antibody directed against BRD9. The fold enrichment of immunoprecipitation with anti-BRD9 antibody compared with nonspecific IgG for some already-known targets of BRD9 [30] was initially analyzed. As shown in Figure 6 (panel A), we demonstrated that the analyzed genes were confirmed to be BRD9 targets in our model. Treatment with GSK2801 displaced the binding of BRD9 to the targets tested, resulting in a significant decrease in enrichment (BRD9 vs. IgG) in most of them (Figure 6, panel B). Among others, *MYCN* was shown to be a target of BRD9 and treatment with GSK2801 was able to disrupt this interaction.

### 2.5. Effects of GSK2801 on MYCN RNA and Protein Levels in SW1736-PTX and 8050C-PTX Cell Lines

After showing the *MYCN* expression reduction by treatment with GSK2801 (Table 1) and considering that *MYCN* is a specific target of BRD9, the effects of such treatment on its protein levels were investigated. First, the decrease in *MYCN* RNA following treatment with GSK2801 20µM was reconfirmed, as shown in Figure 7 (panel A). Data obtained fit with what was observed by transcriptomics analysis. In addition, when evaluating MYCN protein expression by Western blot analysis, GSK2801 treatments induce a significant reduction in both cell lines (Figure 7, panel B and C).

### 2.6. Effects of VPC-70619 on ATC Cell Viability and Colony Formation

Considering the relevance of MYCN in the synergistic mechanism of action of GSK2801 and paclitaxel, we evaluated whether a specific small molecule inhibitor of MYCN (VPC-70619) [31] would be able to alter the observed effects. First, the effect of MYCN inhibition was tested in terms of change in gene expression of its targets that were already described in previous paper [32]. Among the numerous targets described, a few were arbitrarily chosen whose expression would also result from RNA-seq in the cell lines under investigation. Figure 8, panel A, shows the effects of treatment with VPC-70619 on the expression of *MYCN* mRNAs itself, *MDM2*, *CRABP2*, *TGM2*, and *INHBA* in SW1736-PTX and 8505C-PTX cells. *MYCN, MDM2*, and *CRABP2* RNAs were upregulated by treatment with MYCN inhibitor whereas *TGM2* and *INHBA* show a reduction in expression after administration of VPC-70619. Then, we tested VPC-70619 effect on cell viability, alone (Figure 8 panel B) and then in combination with different doses of paclitaxel (Figure 8 panel C). As shown, VPC-70619 determines a significant decrease in cell viability only in concentration higher than 10 μM. However, although using lower doses of VPC-70619, this reduction is most significant when used in combination with significantly lower doses of paclitaxel than those used to make and maintain them resistant. For the following experiments, cells were treated with a combination of the lowest VPC-70619 concentration and showing greater effect on cell viability (PTX 500 nM and VPC-70619 2 μM).

Then, we investigated effects of VPC-70619 on the ability of tumor cells to form colonies in plate. For this reason, we performed a colony formation assay to evaluate whether there were differences between the number of colonies formed after treatment with paclitaxel alone, VPC-70619 alone, or with a combined treatment. The results are reported in Figure 9. As shown, the number of colonies formed after treatment with VPC-70619 decreases significantly in both cell lines. The effect of treatment with VPC-70619 alone was amplified when used in combination with paclitaxel in both cell lines.

### 2.7. Effects of VPC-70619 on In Vitro Tumor Cell Aggressiveness

Since the effects of GSK2801 on in vitro tumor aggressiveness, we repeated the analysis after MYCN inhibition by VPC-70619. VPC-70619 showed an inhibitory effect on soft agar colony formation already in single treatment but even more in combination treatment in both cell lines. The data are shown in Figure 10, panel A (8505C-PTX) and panel B (SW1736-PTX), and illustrate a similarity with those obtained in the previous experiment since again, the combined treatment has a greater inhibitory effect on colony-forming ability. Wound closure assay demonstrated that, in both cell lines, treatment with paclitaxel alone led to nearly complete wound closure (higher wound closure rate). Compared with this control situation, single treatment with VPC-70619 does not result in wound closure (Figure 10, panels C and D). This effect is more evident in the case of combined treatment; this means that VPC-70619 has a higher inhibitory effect on cell migratory capacity when used in combination with paclitaxel.

## 3. Discussion

Whereas the combination of thyroid gland surgical excision and involved lymph nodes as well as irradiation with radioactive iodine (RAI) is effective for most thyroid tumors, this approach is not effective in benefiting the treatment of poorly differentiated thyroid carcinoma or ATC [33]. For this subclass of thyroid neoplasms showing such a high degree of malignancy, there are numerous therapeutic approaches that have been investigated in recent years [34,35,36]. However, identifying the right strategy to improve the life expectancy of ATC patients still remains a challenging task, partly because of the development of resistance to therapy. Our research group has been moving for years in the field of studying new therapeutic agents that can cope with the progression of ATC [37]. After experimenting with natural compounds, among others, we generated two cell lines resistant to paclitaxel treatment, derived from two anaplastic thyroid cancer cell lines (SW1736 and 8505C) [3]. Paclitaxel indeed, is a taxane still indicated in mono- or multi-therapy for those tumors in which the molecular signature does not suggest the use of personalized treatment with a tyrosine kinase inhibitor [4]. In our previous studies, we obtained interesting results regarding the gene expression profile of paclitaxel-resistant ATC cells compared with susceptible cells; moreover, we had already demonstrated the role of bromodomain and extra-terminal (BET) proteins and their inhibitors in this tumor type [3,38,39]. Therefore, we focused our attention on three genes coding for bromodomain-containing proteins (BAZ2A, BAZ2B, and BRD9) that were all expressed in paclitaxel-resistant SW1736 and 8505C cancer cell lines. First, we analyzed the expression of the three target genes demonstrating overexpression of BRD9 in paclitaxel-resistant cells, with no significant effect on the expression of BAZ2A and BAZ2B (Figure 1). Based on the increased expression of BRD9 and the importance of bromodomain-containing proteins in thyroid tumor development, tumor growth, and progression, we tested the efficacy of a specific inhibitor of these three proteins (GSK2801) [25]. In the literature, it has already been described that GSK2801 alone does not have particular effectiveness on triple-negative breast cancer cells, while its effectiveness increases sharply when used synergistically with another inhibitor [26]. GSK2801 showed no effect on cell viability of paclitaxel-sensitive SW1736 and 8505C cell lines, even when used at high doses (20 µM, 40 µM). When it was tested on paclitaxel-resistant ATC cells, consistently with what was observed in the literature, the administration of GSK2801 alone does not cause any particular effects, whereas in association with paclitaxel, important and statistically significant effects in terms of cell viability were observed (Figure 2). This effect would be due to partial GSK2801-mediated resistant cell paclitaxel resensitization. Although further investigation into the mechanisms underlying the reduction in cell viability is needed, part of the phenomenon could be explained by the increase in cell death by apoptosis. Resistant paclitaxel ATC cells also showed a reduced ability to form plate colonies when treated with a combination of GSK2801 and paclitaxel (Figure 3). In addition to encouraging data on cell proliferation, the combined treatment showed great effectiveness in reducing the in vitro aggressiveness behavior, a feature evaluated by investigating the ability of cells to form colonies in soft agar or their ability to migrate. Again, GSK2801 when used alone did not appear to have significant effects, suggesting a possible reversion of resistance given the fact that it showed its effects only when used in combination with paclitaxel. In this study, we firstly demonstrated that the inhibition of BAZ2A, BAZ2B, and BRD9 protein activity by a specific inhibitor is able to strongly reduce the viability, to induce apoptosis, and to decrease clonogenic capacity of resistant paclitaxel cells when used in synergism. Furthermore, these effects also extended to in vitro tumor aggressiveness, which appeared significantly reduced in SW1736-PTX and 8505C-PTX cells when GSK2801 was co-administered with paclitaxel. Taken together, these results suggest that, since the action of GSK2801 appears significant only in synergy with paclitaxel, the mechanism of action may relate to partial resensitization of resistant cells. Therefore, it is possible to hypothesize that the bromodomain proteins inhibited by GSK2801 are no longer able to fulfill their function as regulators of gene transcription, thus altering the pattern of gene expression necessary to maintain resistance to paclitaxel treatment in these cells. In light of the marked biological effects of synergistic treatment, we then focused on the molecular mechanisms by which GSK2801 might alter the balance of ATC cells not sensitive to paclitaxel, making them susceptible to such treatment again. To this purpose, the effects of GSK2801 treatment on gene expression were investigated. To assess the most direct effects, an RNA-seq analysis was performed on RNA extracted from cells treated with GSK2801 since it is a competitive inhibitor that prevents its targets from interacting with proteins-DNA; its action does not have to go through, for example, downregulation of the expression of BAZ2A, BAZ2B, and BRD9 proteins. Since it immediately acts on the already available proteins, its effects can be presumed to be measurable as early as after initial treatment, when on the other hand, there is not yet an alteration in gene expression induced by cellular suffering secondary to treatment. In contrast to the number of genes with altered expression after treatment with GSK2801 in either cell line, the number of dysregulated genes commonly in SW1736-PTX and 8505C-PTX cells was more modest (Figure 5). This is probably attributable to the fact that the two cell lines have different expression profiles, but the genes that respond to treatment are presumably few and specific. In light of this, the hypothesis that the effects of GSK2801 go through resensitization to paclitaxel of resistant cells finds additional support, as resistance to this taxane is an acquired characteristic shared by two different cell lines. Assuming a greater linearity of effect between treatment and downregulation of gene expression, genes in both cell lines underexpressed relative to controls were investigated. After evaluating the above genes by basal expression level, cell function, and relevance in tumorigenesis, the focus was on MYCN. MYC family members, MYC, MYCN, and MYCL, are transcription factors that regulate the expression of genes involved in normal development, cell growth, proliferation, metabolism, and survival. MYC dysregulation appeared to be tissue-specific tissue during tumorigenesis; MYCN was described to be aberrantly expressed in neuronal or neuroendocrine tumors, and more recently, in ovarian cancer, prostate cancer, and in a subset of breast cancer with poor prognosis [40,41,42]. The emergence in our analysis of MYCN as a likely most relevant target following treatment with GSK2801 would also be strongly supported by extensive literature on this topic, which shows that there is a strong correlation between overexpression of MYCN and response to treatment with BET inhibitors used alone or in combination [43,44]. *MYCN* turned out to be a direct target of BRD9 in this cell model (Figure 6), and inhibition of MYCN-target binding acted by GSK2801 determined a marked and significant reduction in MYCN protein levels in SW1736-PTX and 8505C-PTX cells (Figure 7). Based on the hypothesis that MYCN was one of the main actors downstream of the biological effects of GSK2801, a specific MYCN inhibitor was tested to assess whether it could mimic the effects of a MYCN downregulation. SW1736-PTX and 8505C-PTX cells were treated with VPC-70619, a small molecule inhibitory MYCN [31], at a concentration at which, when used alone, it was unable to determine effects on cell viability (2 μM). VPC-70619 also demonstrated its specificity of direct action against MYCN in paclitaxel-resistant SW1736 and 8505C cells, as was shown by the change in expression levels of some of its targets (Figure 8). MYCN itself, MDM2, and CRABP2 were upregulated after MYCN inhibition, while TGM2 and INHBA were downregulated, in agreement with what was already observed by Bell et al. [32]. Similar to that observed in co-treatment with GSK2801 also in the case of simultaneous administration of VPC-70619 and paclitaxel, a strong reduction in cell viability was observed, at concentrations at which neither drug had an effect individually (Figure 8). The effective dose of VPC-70619, when used in cotreatment with paclitaxel, was 10 times lower than that of GSK2801 (2 μM vs. 20 μM). This could be due to the fact that BRD9 acts as a modulator of gene expression of a plethora of genes; its inhibition therefore induces fewer direct effects than specific suppression of MYCN. Even in this case, co-treatment of VPC-70619 and paclitaxel showed inhibition of plate colony formation and reduction of tumor aggression, as demonstrated by soft agar assay and wound healing assay. In conclusion, inhibition of MYCN activity by VPC-70619 showed significant biological effects on paclitaxel-resistant ATC cell lines when used together with paclitaxel. This suggests that the lack of functionality of MYCN may result in partial resensitization of the cells examined and, ultimately, that a substantial part of the effect of GSK2801 comes through inhibition of MYCN expression. This preliminary study opens a way to understanding the mechanisms of how BET proteins exert their role in anaplastic thyroid cancer and how their inhibition may represent a therapeutic target for the treatment of a type of tumor which is to date still hardly curable. In addition, these results offer a molecular explanation of the observed effects by applying MYCN to high-profile targets in this type of mechanism of action. As far as we know, this is the first study in which the effects of BAZ2A, BAZ2B, BRD9, and MYCN inhibition in paclitaxel-resistant anaplastic thyroid carcinoma cell lines, are examined, providing a starting point for further and more in-depth studies that can fill the gap of effective treatments for this subcategory of thyroid tumors.

## 4. Materials and Methods

**Cell lines** In this study, we used two thyroid cancer cell lines derived from anaplastic thyroid cancer, SW1736 and 8505C, and two cell lines derived from the latter that acquired resistance to paclitaxel: SW1736-PTX and 8505C-PTX. All cell lines were cultured in RPMI-1640 medium (Euroclone S.p.A, Milano, Italy) supplemented with 10% FBS (Gibco; Thermo Fisher Scientific, Inc., Waltham, MA, USA), 2 mM L-glutamine (Euroclone S.p.A, Milano, Italy), and 50 mg/mL gentamicin (Gibco; Thermo Fisher Scientific, Inc.). Cells were cultured in a humidified incubator (5% CO_2_ and 95% air at 37 °C) (Eppendorf AG, Hamburg, Germany). We obtained ATC paclitaxel-resistant cell lines as previously described [3]. Resistant cells were cultured in RPMI-1640 medium supplemented with 10%, 2 mM L-glutamine, 50 mg/mL gentamicin, and paclitaxel 1.5 μM. In this manuscript, we will refer to them as SW1736-PTX and 8505C-PTX. SW1736-PTX and 8505C-PTX cells were treated with paclitaxel (Selleck Chemicals, Houston, USA), GSK2801 (Selleck Chemicals Houston, USA) and VPC-70619 (MedChem Express NJ, USA).

**Cell viability** In order to test cell viability, the MTT assay was used, as previously described [34]. SW1736-PTX and 8505C-PTX cells were seeded in 96-well plates (4 × 10^3^ cells/well). The following day, cells were treated with PTX or GSK2801 at different concentrations and different combinations. After 72 h of incubation, 4 mg/mL MTT (Sigma-Aldrich, Darmstadt, Germany; Merck KGaA, Darmstadt, Germany) was added to the cell medium and cells were cultured for a further 4 h in the incubator. The supernatant was removed, 100 µL/well DMSO was added, and the absorbance at 570 nm was measured. The same experiment was performed with paclitaxel or VPC-70619 treatment, following the same protocol. All experiments were performed as six technical repeats and cell viability is expressed as the fold-change relative to the control. EC50 was determined from the dose-response curves using the percentage of cell viability.

**Colony formation assay** The clonogenic activity of the paclitaxel-resistant ATC cell lines was evaluated by colony formation assay. Briefly, SW1736-PTX and 8505C-PTX cells were treated with paclitaxel, GSK2801, or in combination and then the cells were seeded in 10-cm plates at a density of 1000 cells/plate. Adherent colonies were fixed using methanol and then stained with crystal violet colorimetric dye, which allows for visual quantification of the number of colonies that expanded. The same experiment was performed with paclitaxel or VPC-70619 treatment, following the same protocol.

**Soft agar assay** The clonogenic ability of the SW1736-PTX and 8505C-PTX cells was studied using a soft agar assay. After 48 h of treatment, cells were collected and 20,000 cells were suspended in complete medium containing 0.25% agarose (PanReac AppliChem ITW Reagents, Darmstadt, Germany), then seeded to the top of a 1% agarose complete medium layer in 6 cm plates. The colonies were counted in four different fields under a Leica DMI-600B inverted microscope (Leica Microsystems Ltd., Wetzlar, Germany). The same experiment was performed with paclitaxel or VPC-70619 treatment, following the same protocol. Data are representative of three independent experiments.

**Wound healing assay** The migration capability of the cells was evaluated by wound healing assay. The cells were seeded into 6-well plates (500,000 cells/well) and the day after they were treated with paclitaxel, GSK2801, or with a combined treatment, or in the second case, they were treated with paclitaxel, VPC-70619, or with a combined treatment. A linear scratch was performed with a 200 μL sterile pipette tip across the cell monolayer. Cells were then incubated in a humidified incubator at 37 °C and images of the scratched monolayer were acquired after 0, 6, 24, and 30 h with an inverted microscope Leica DMI-600B (Leica Microsystems Ltd.). Experiments were run in triplicate.

**RNA Extraction and RNA quantification** Total RNA from SW1736, 8505C, SW1736-PTX, and 8505C-PTX cells was extracted using RNeasy Mini Kit according to the manufacturer’s instructions (Qiagen, Hilden, Germany). In order to quantify the RNA through Qubit 4.0 Fluorometer, the Qubit RNA HS assay (ThermoFisher Scientific, Waltham, MA, USA) was used.

**Gene expression assays** A total of 1 μg total RNA from all cell lines was extracted as described above and reverse transcribed to cDNA using random hexaprimers and SuperScript IV reverse transcriptase (Thermo Fisher Scientific, Waltham, MA, USA). Quantitative PCR was performed using PowerUP Sybr green master mix (Thermo Fisher Scientific, Waltham, MA, USA) on the QuantStudio3 system (Applied Biosystems, Waltham, MA, USA). The QuantStudio Design and Analysis software v1.5.0 (Applied Biosystems) were used to calculate mRNA levels with the 2^−ΔΔCt^ method, and ß-actin was used as a reference. All experiments were performed in triplicate. Oligonucleotide primers for ß-actin (forward 5′-TTGTTACAGGAAGTCCCTTGCC-3′ and reverse 5′-ATGCTATCACCTCCCCTGTGTG-3′), BAZ2A (forward 5′-AAGATGTGTGGCTACAATGG-3′ and reverse 5′-TCTGCACCCATCAGCTCCG-3′), BAZ2B (forward 5′-ACCCCCAAACCAAATGCTGGTGC-3′ and reverse 5′-TGCTGTTGACTGAACGCTGCCC-3′), BRD9 (forward 5′-ATGTTCCATGAAGCCTCCAG-3′ and reverse 5′-AGCTCCTTCTTCACCTTCCC-3′), CRABP2 (forward 5′-GCCTGGTGAAATGGGAGAGT-3′ and reverse 5′- CAGTTCCCCATCGTTGGTCA-3′), INHBA (forward 5′-GCTTCATGTGGGCAAAGTCG-3′ and reverse 5′- TGACTCGGCAAACGTGATGA-3′), MDM2 (forward 5′-TGGCCAGTATATTATGACTAAACGA-3′ and reverse 5′- TCACAGAGAAGCTTGGCACG-3′), MYCN (forward 5′-AATCGACGTGGTCACTGTGG-3′ and reverse 5′-GACCCAGGGCTGCGTTCTT-3′), TGM2 (forward 5′-TGATCTGGAGCTGGAGACCA-3′ and reverse 5′- TACACTGGCCTCGTAGTTGC-3′) were purchased from Sigma-Aldrich (Darmstadt, Germany).

**RNA-sequencing** After treating the cells according to the described experimental plan for 6 h, RNA was extracted and subjected to RNA-sequencing. Barcoded libraries were prepared using the Ion AmpliSeq Transcriptome Panel Human Gene Expression CORE (Thermo Fisher Scientific, Waltham, MA, USA) and the Ion AmpliSeq Library Kit Plus (Thermo Fisher Scientific, Waltham, MA, USA), following the manufacturer’s protocol. Veriti Dx 96-Well Thermal Cycler (Applied Biosystems, Waltham, MA, USA) was used for all reactions. Barcoded libraries were quantified with the Qubit dsDNA HS Assay kit (Life Technologies, Carlsbad, CA, USA) and then diluted to 100 pM. Libraries were loaded into the Ion Chef instrument (Thermo Fisher Scientific, Waltham, MA, USA) for template enrichment and chip loading. Sequencing was performed with the Ion S5 GeneStudio Sequencer using the Ion 540 Kit-Chef and the Ion 540 chip-kit (all Thermo Fisher Scientific, Waltham, MA, USA). Reads were aligned to the reference genome and the RNA-seq analysis plugin was run on the Torrent Suite Server (Thermo Fisher Scientific, Waltham, MA, USA). For further analysis, we selected effective data for those with FPKM values >0.5 and log2 fold-change >1.5 or <−1.5. In the evaluation of commonly up or downregulated genes, the cut-offs were increased to −3 and +3 in order to obtain more representative results. Four biological replicates were made for each experimental point and then pulled together.

**Protein Extraction and Western Blot** For protein collection, extraction and later use for Western blot, the previously described method was followed [45]. Proteins were run on 7.5% SDS–PAGE and then transferred to nitrocellulose membranes (GE Healthcare, Little Chalfont, UK), saturated with 5% non-fat dry milk in PBS/0.1% Tween 20. The membranes were incubated overnight with rabbit polyclonal anti-cleaved-PARP antibody 1:1000 (Invitrogen), rabbit anti-actin antibody 1:1000 (Merck KGaA), rabbit anti-BAZ2A antibody 1:1000 (Abcam, Cambridge, United Kingdom), rabbit anti-BAZ2B antibody 1:1000 (Abcam, Cambridge, United Kingdom), mouse anti-MYCN antibody 1:500 (Santa Cruz Biotechnology, Inc., Dallas, TX, USA), and rabbit anti-BRD9 antibody 1:1000 (Bethyl, Fortis Life Sciences, Waltham, MA, USA). The day after, membranes were incubated with anti-rabbit or anti-mouse immunoglobulin coupled to peroxidase 1:8000 (Merck KGaA) for 2 h. Blots were developed using UVITEC Alliance LD (UVITec Limited, Cambridge, UK) with the SuperSignal Technology (Thermo Scientific Inc. Waltham, MA, USA).

**Chromatin immunoprecipitation (ChIP)** The ChIP assay was performed as previously described [46]. For immunoprecipitation, samples were incubated with 10 μg of rabbit polyclonal anti-BRD9 (Bethyl, Fortis Life Sciences) or Rabbit IgG, (Millipore, Burlington, MA, USA), as negative control. After incubation with Dynabeads Protein A, washing, and elution, the DNA was extracted with phenol/chloroform; the aqueous phase was recovered and precipitated. The extracted DNA was used as a template in qPCR with the following primers: MYCN (forward 5′-AATCGACGTGGTCACTGTGG-3′ and reverse 5′-GACCCAGGGCTGCGTTCTT-3′), E2F1 (forward 5′-CTAGGTTCTGCTCCCGACAG-3′ and reverse 5′- CCTCAAGGTCCAACTCCAGC-3′), ELK4 (forward 5′- TGTGATGTTCCCCTGTGCTC-3′ and reverse 5′- TGCTTCTCTGCGGCCTAAAA-3′), MYO7 (forward 5′- CCCAAACAGCCAGGTAGAGG-3′ and reverse 5′- CTGTTGTTTGCAGGACACCG-3′) and NEF2L1 (forward 5′- TGGCTATGGTATCCACCCCA-3′ and reverse 5′- ACCAGCCAGGCATTTACCTC-3′). Fold enrichment was then calculated as signal over background (IgG).

**Statistical analysis** All data obtained are expressed as means ± standard deviation, and significances were analyzed with the student’s t-test performed with GraphPAD Software for Science (San Diego, CA, USA).

## Figures and Tables

**Figure 1 ijms-24-05993-f001:**
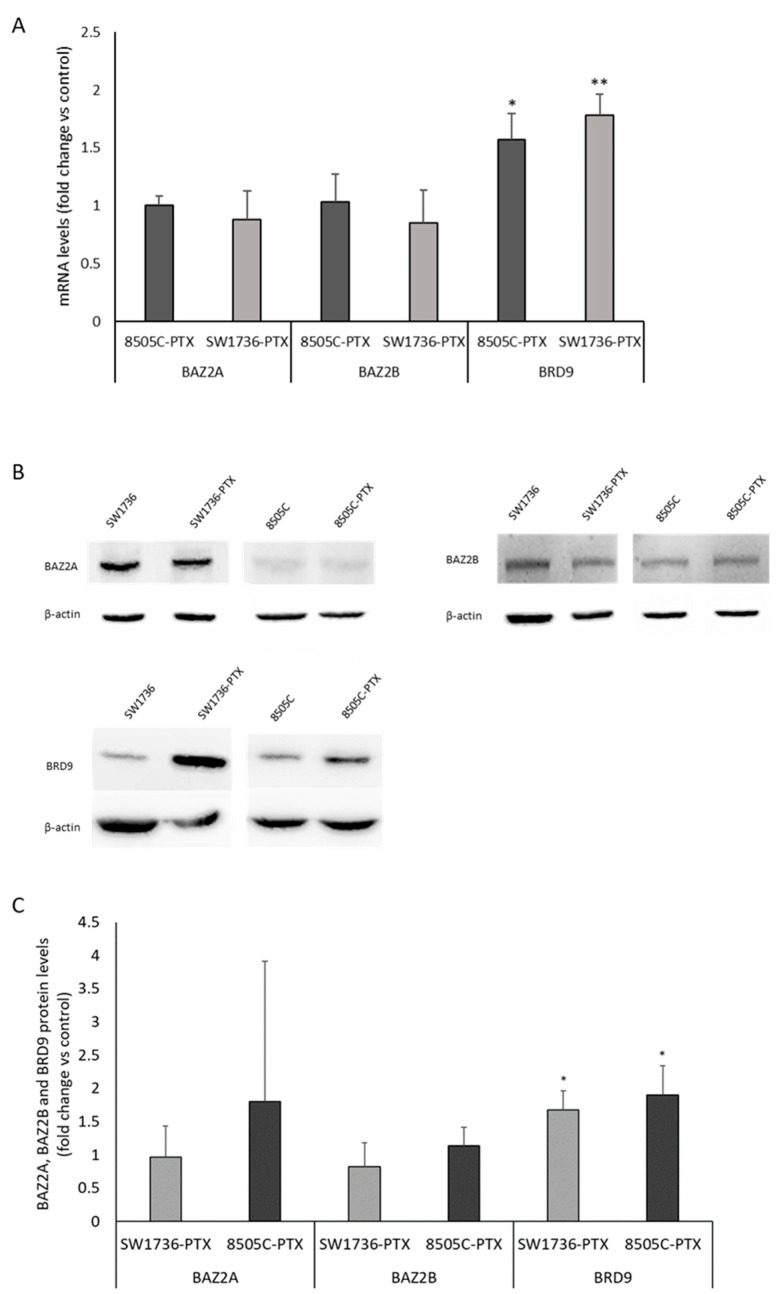
*BAZ2A*, *BAZ2B*, and *BRD9* gene expression and protein levels. (**A**) Relative expression levels of *BAZ2A*, *BAZ2B*, and *BRD9* mRNA in 8505C-PTX and SW1739-PTX. For each cell line, mRNA expression in the paclitaxel-sensitive parental cell line was set at 1. (**B**) Western blot analysis of BAZ2A, BAZ2B and BRD9 in paclitaxel-sensitive anaplastic thyroid cancer cell lines (SW1736 and 8505C), SW1736-PTX and 8505C-PTX cells. (**C**) Densitometric analysis of BAZ2A, BAZ2B and BRD9 protein levels in these four cell lines. For each cell line, the results were normalized against β-actin levels and expressed in arbitrary unit. Protein expression in the paclitaxel-sensitive parental cell line was set at 1 and data are expressed as fold change respect to control. n = 3, * *p* < 0.05, ** *p* < 0.01.

**Figure 2 ijms-24-05993-f002:**
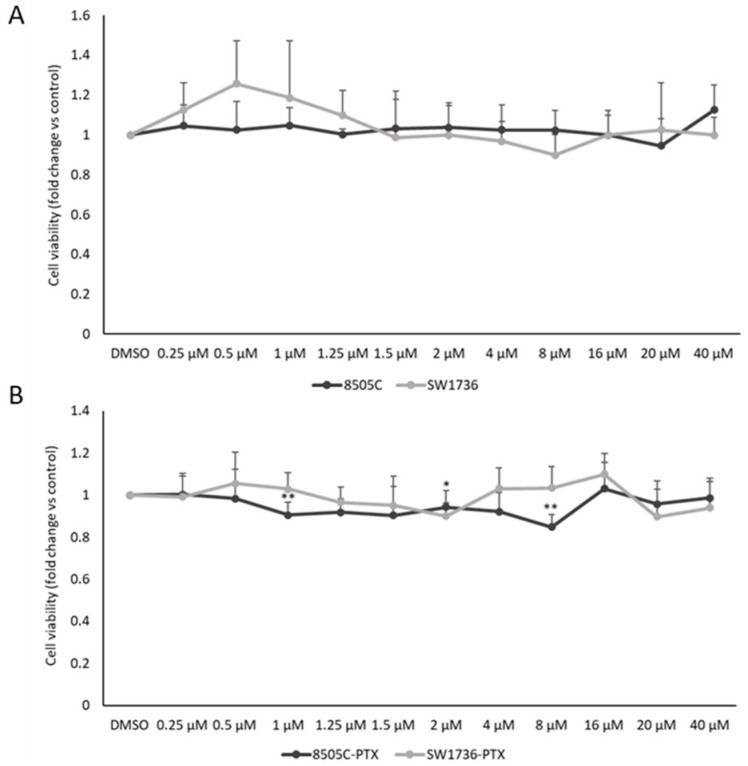
Effect of GSK2801 on cell viability. (**A**) Both 8505C and SW1736 cells were treated with GSK2801 at different doses (rising from 0.25 μM to 40 μM) or vehicle (DMSO) for 72 h and cell viability was assessed by MTT assay. (**B**) Both 8505C and SW1736 paclitaxel-resistant (8505C-PTX and SW1736-PTX) cells were treated with GSK2801 at different doses (rising from 0.25 μM to 40 μM) or vehicle (DMSO) for 72 h and cell viability was assessed by MTT assay. (**C**) Both 8505C-PTX and SW1736-PTX were treated with paclitaxel alone (PTX), GSK2801 alone, or in different combination with paclitaxel for 72 h. Cell viability was evaluated by MTT assay. Each point represents the mean of six measurements. n = 6. p values refer to the comparison of GSK2801 alone or in combination. (**D**,**E**) Cleaved-PARP protein levels were evaluated as apoptosis marker by Western blot assay in SW1736-PTX and 8505C-PTX cells treated with vehicle (PTX) or paclitaxel 200 nm and GSK2801 20 μM (GSK2801). For each cell line, the results were normalized against β-actin levels and expressed in arbitrary unit. Data are representative of 3 independent experiments and results are shown as mean ± SD. * *p* < 0.05, ** *p* < 0.01, *** *p* < 0.001, **** *p* < 0.0001.

**Figure 3 ijms-24-05993-f003:**
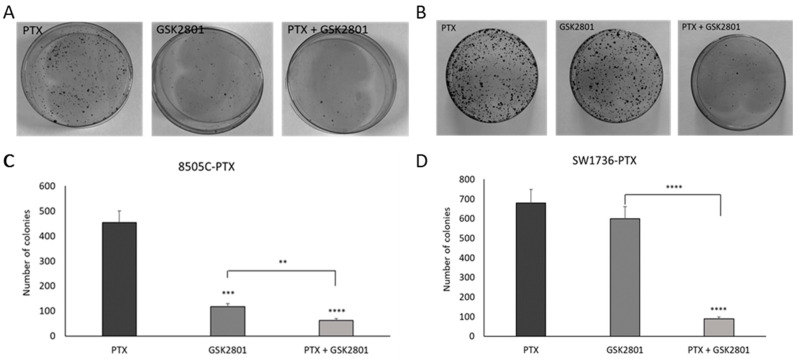
Effects of GSK2801 treatment on colony forming ability in resistant ATC cell lines. The ability to form colonies of these cell lines following single treatment with paclitaxel (PTX), GSK2801 alone (GSK2801), or in combination with paclitaxel (PTX + GSK2801) was studied by colony formation assay. Panels (**A**,**C**) represent 8505C-PTX cells, while panels (**B**,**D**) represent SW1736-PTX cells. In panels (**C**,**D**), histograms represent the number of colonies. n = 3. ** *p* < 0.01, *** *p* < 0.001, **** *p* < 0.0001.

**Figure 4 ijms-24-05993-f004:**
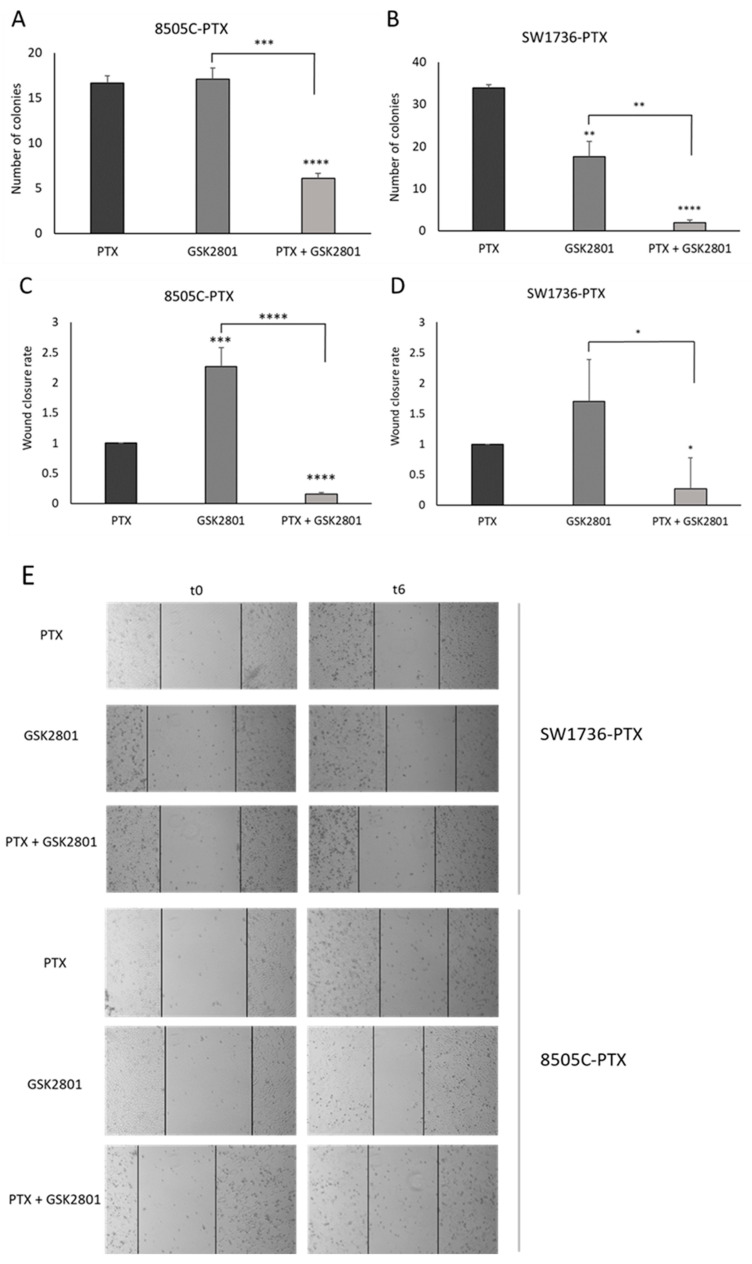
Effects of GSK2801 treatment on the aggressive behavior of cancer cells. The effect of GSK2801 on the ability to form colonies in an anchorage-independent way was measured by soft agar assay. The data obtained in 8505C-PTX cells are shown on the left panel (**A**), while on the right panel (**B**) the data obtained in SW1736-PTX cells are shown, following single treatment (paclitaxel alone here reported as PTX and GSK2801 alone) or combined treatment (PTX + GSK2801). The histograms represent the number of colonies per cell line. Data are representative of 3 independent experiments. (**C**,**D**) Evaluation of the ability of cells to migrate under the same conditions of soft agar assay, data obtained by wound healing assay. The wound closure rate was calculated measuring the difference in wound width (WW) at time zero (t0) and after 6 h (t6). The formula used is as follows: Wound closure rate = (*WWt*0 − *WWt*6) *treatment* ÷ (*WWt*0 − *WWt*6) *control* n = 4. (**E**) Photos showing wound margins (continuous black lines) at the beginning of migration (t0) and after 6 h (t6). * *p* < 0.05, ** *p* < 0.01, *** *p* < 0.001, **** *p* < 0.0001.

**Figure 5 ijms-24-05993-f005:**
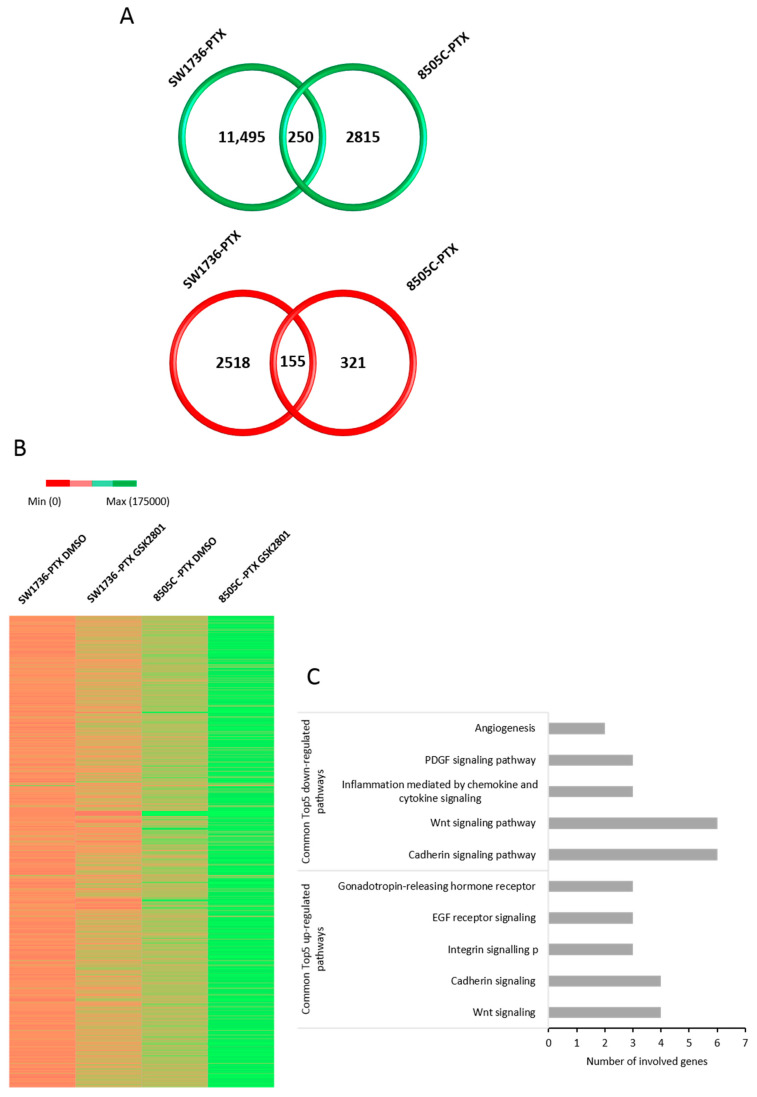
Effects of GSK2801 on gene expression. Venn diagrams represented the comparison of upregulated (green) and downregulated (red) between SW1736-PTX and 8505C-PTX cell lines treated with 20 μM GSK2801, after RNA-seq data analysis. Within the intersection of the circles are indicated the shared modified genes between the two cell lines panel (**A**). Panel (**B**): heat maps obtained after gene expression analysis following treatment with GSK2801 (SW1736-PTX GSK2801 and 8505C-PTX GSK2801) or with the vehicle alone (SW1736-PTX DMSO and 8505C-PTX DMSO) in ATC cells. Panel (**C**): top5 down- and upregulated pathways common in the two cell lines after GSK2801 treatment (20 μM for 6 h).

**Figure 6 ijms-24-05993-f006:**
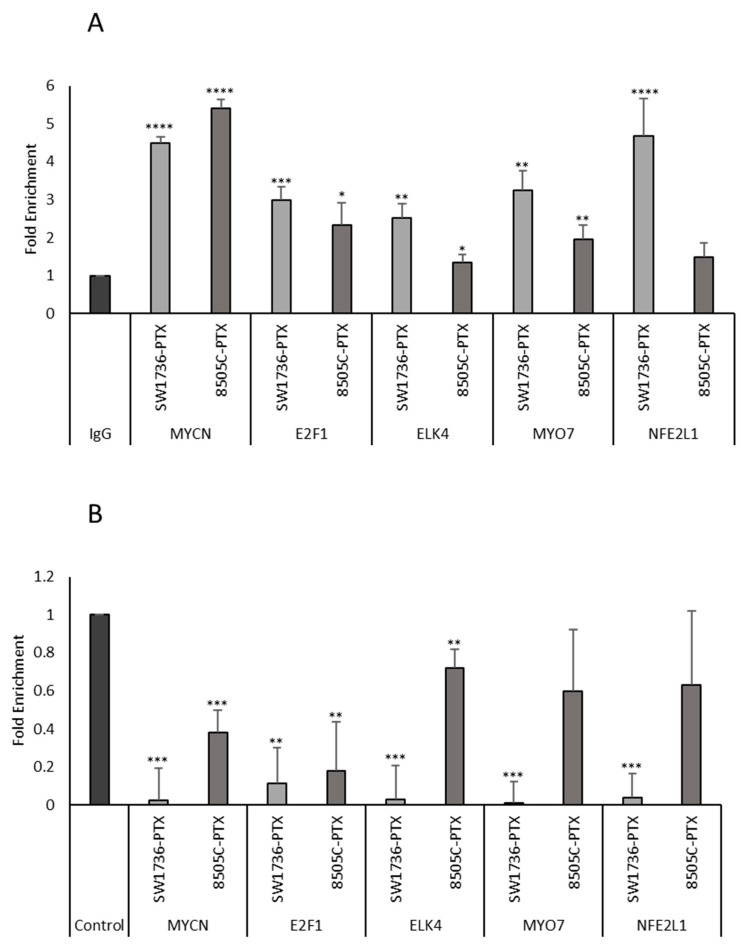
Chromatin was isolated from SW1736-PTX and 8505C-PTX cell lines and immunoprecipitated with BRD9 or IgG antibodies. *MYCN*, E2F1, ELK4, MYO7, NFE2L1 DNAs were amplified using promoter-specific primers and analyzed by qPCR. All samples were run in triplicate. IgG-immunoprecipitate was arbitrarily set at 1.0 and the enrichment was expressed as relative expression value panel (**A**). SW1736-PTX and 8505C-PTX cells were treated with GSK2801 20 µM or with DMSO only (control). After chromatin was immunoprecipitated with BRD9 or IgG antibodies, *MYCN*, E2F1, ELK4, MYO7, NFE2L1 DNAs were amplified using promoter-specific primers and analyzed by qPCR panel (**B**). All samples were run in triplicate. Enrichment versus IgG of control (cells not treated with GSK2801) was arbitrarily set at 1 (control). n = 3. * *p* < 0.05, ** *p* < 0.01, *** *p* < 0.001, **** *p* < 0.0001.

**Figure 7 ijms-24-05993-f007:**
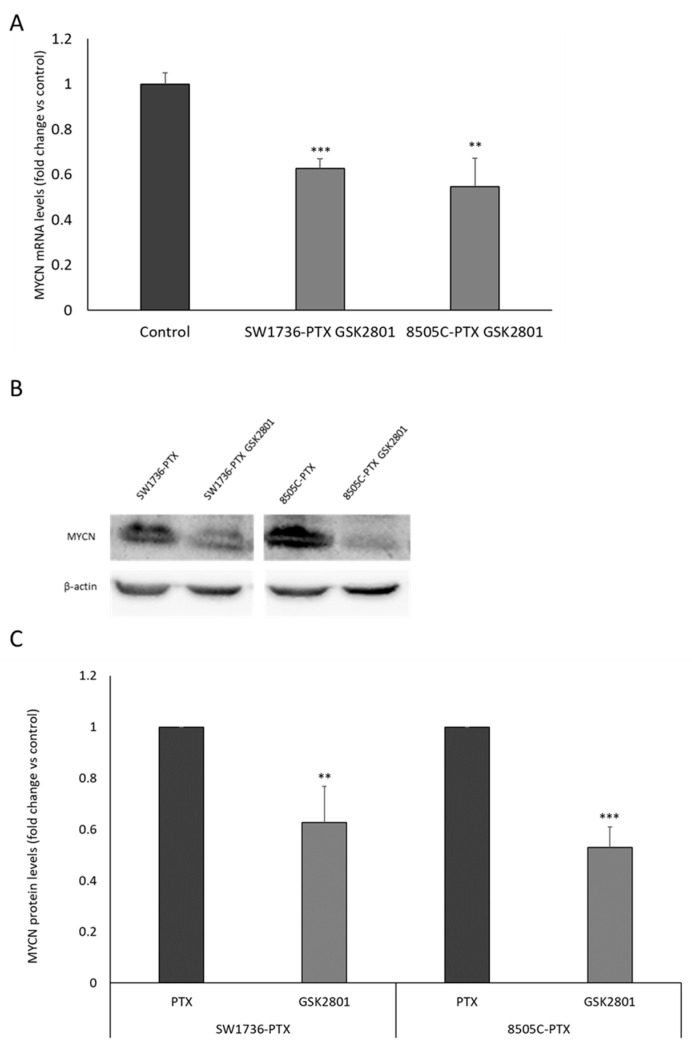
Effects of GSK2801 on *MYCN* RNA and protein levels. Panel (**A**): Relative expression levels of *MYCN* mRNA in SW1736-PTX and 8505C-PTX treated with GSK2801 20 µM (SW1736-PTX GSK2801 and 8505C-PTX) or with DMSO only (control). All samples were run in triplicate. Control was arbitrarily set at 1.0. Panel (**B**) Western blot analysis of MYCN protein level in SW1736-PTX or 8505C-PTX treated with GSK2801 20 µM (SW1736-PTX GSK2801 and 8505C-PTX GSK2801) or not (SW1736-PTX or 8505C-PTX). Panel (**C**): Densitometric analysis of MYCN protein levels in SW1736-PTX and 8505C-PTX cells treated with GSK2801 (GSK2801) or not (PTX). For each cell line, the results were normalized against β-actin levels and expressed in arbitrary unit. n = 3. ** *p* < 0.01, *** *p* < 0.001.

**Figure 8 ijms-24-05993-f008:**
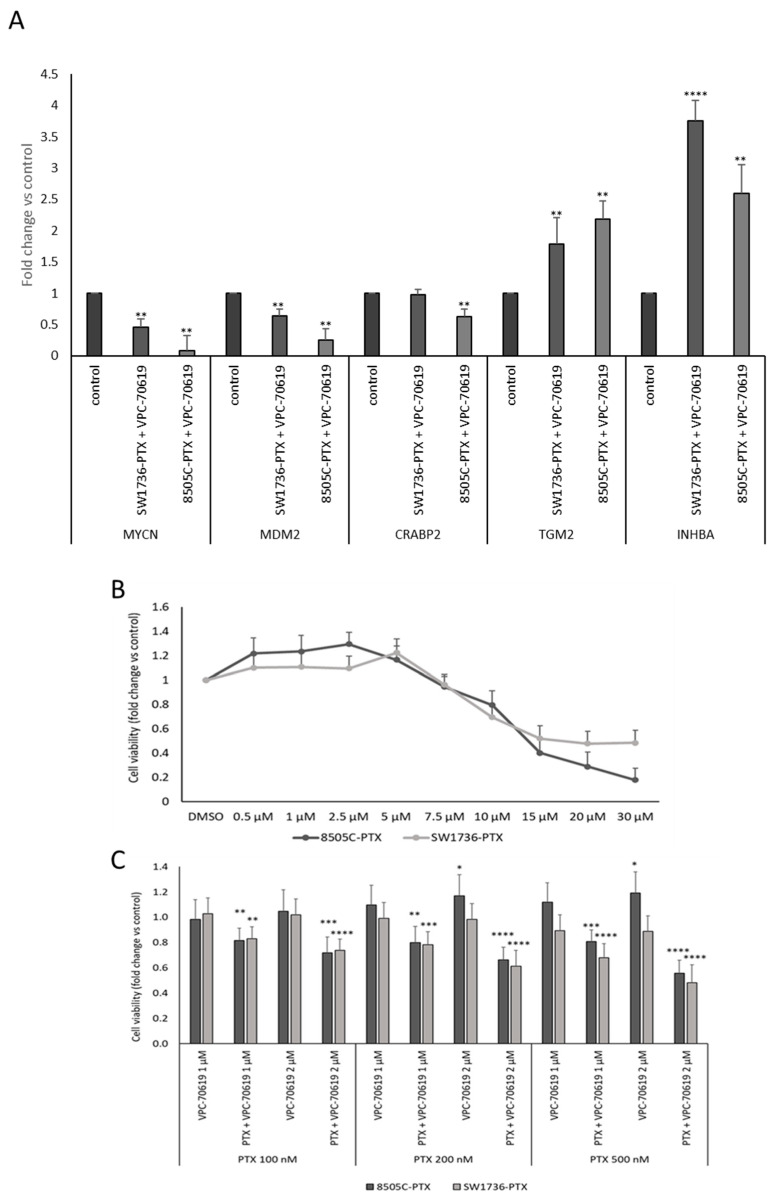
Panel (**A**): SW1736-PTX and 8505C-PTX cells were treated with VPC-70619 (SW1736-PTX + VPC-70619 and 8505C-PTX + VPC-70619) or with DMSO as vehicle (control), after mRNA levels of some MYCN targets were evaluated by real-time PCR after following retro transcription. Control was arbitrarily set at 1.0. n = 3. * *p* < 0.05, ** *p* < 0.01, *** *p* < 0.001, **** *p* < 0.0001. Panel (**B**): 8505C and SW1736 paclitaxel-resistant (8505C-PTX and SW1736-PTX) cells were treated with VPC-70619 at different doses (rising from 0.5 μM to 30 μM) or vehicle (DMSO) for 72 h and cell viability was assessed by MTT assay. Panel (**C**): 8505C-PTX and SW1736-PTX were treated with VPC-70619 at two different concentration (1 or 2 μM) in combination with paclitaxel (PTX) at three different concentrations (100, 200, or 500 nM) for 72 h. Cell viability data are obtained by MTT assay. Each point represents the mean of six measurements. n = 6. Panel (**A**) *p*-values are listed in Appendix A.

**Figure 9 ijms-24-05993-f009:**
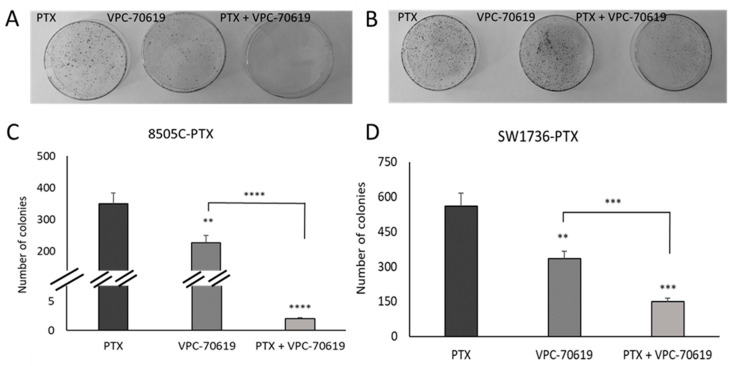
Effects of VPC-70619 treatment on colony-forming ability in resistant ATC cell lines. The ability to form colonies in an anchorage-dependent way was studied by colony formation assay in both cell lines following single treatment with paclitaxel (PTX), VPC-70619 and in combination with paclitaxel (PTX+VPC-70619). Panels (**A**,**C**) represent 8505C-PTX cells, while panels (**B**,**D**) represent SW1736-PTX cells. In panel C and panel D, histograms represent the number of colonies per cell line. n = 3. ** *p* < 0.01, *** *p* < 0.001, **** *p* < 0.0001.

**Figure 10 ijms-24-05993-f010:**
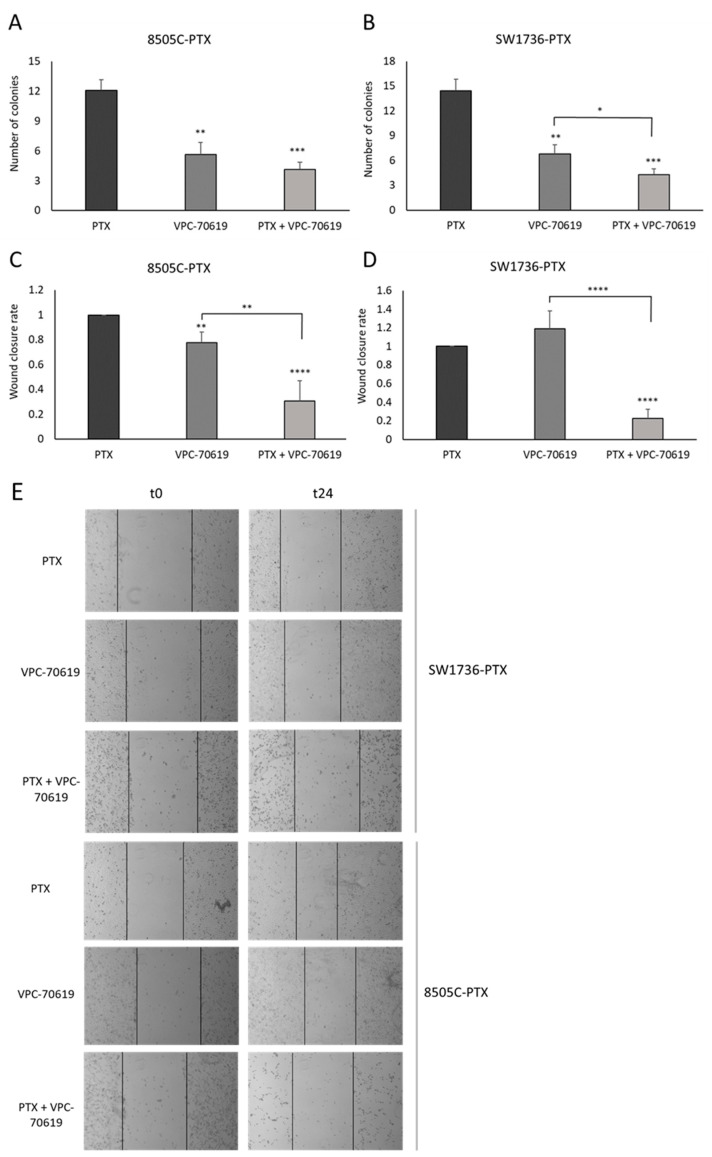
Effects of VPC-70619 treatment on the aggressive behavior of cancer cells. Panel (**A**,**B**): inhibitory effects of VPC-70619 on the ability to form colonies in an anchorage-independent way, which was studied by soft agar assay. The y-axis of each histogram shows the number of colonies formed for each condition (treatment with paclitaxel (PTX), VPC-70619 alone /VPC-70619) and combination (PTX + VPC-70619)). n = 3. Panels (**C**,**D**) show the change in cell migratory capacity after single and combined treatment, in both cell lines. Migratory capacity was evaluated by wound healing assay. The wound closure rate was calculated measuring the difference in wound width (WW) at time zero (t0) and after 6 h (t6). The formula used is as follows: Wound closure rate = (*WWt*0 − *WWt*6) *treatment* ÷ (*WWt*0 − *WWt*6) *control*. Panel (**E**): Photos showing wound margins (continuous black lines) at the beginning of migration (t0) and after 6 h (t6). * *p* < 0.05, ** *p* < 0.01, *** *p* < 0.001, **** *p* < 0.0001.

**Table 1 ijms-24-05993-t001:** List of genes commonly up- or downregulated in SW1736-PTX and 8505C-PTX after GSK2801 treatment and involved in WNT signaling pathway, cadherin signaling pathway, or both.

SW1736-PTX and 8505C-PTX Common Altered Genes	WNT Signaling Only	WNT_Cadherin Signaling Common	Cadherin Signaling Only
Downregulated	*MYCN*	*PCDHGA5, PCDHB4, PCDHGB6, PCDHGC4*	*ACTBL2*
Upregulated	*DCHS1*	*DCHS1, PCDHGA8, PCDHB12*	*ERBB4*

## Data Availability

The datasets generated and/or analyzed during the current study are available from the corresponding author on reasonable request.

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
