# Peer review of "GSK2801 Reverses Paclitaxel Resistance in Anaplastic Thyroid Cancer Cell Lines through MYCN Downregulation"

_ijms, 2023, doi:10.3390/ijms24065993_

Round 1

Reviewer 1 Report

The manuscript is acceptable after minor revision.

1. In the introduction, there is no explanation about combination therapy and its benefits, so it is suggested to use the following references:     DOI: 10.22038/ijbms.2016.6649                                                          DOI: 10.1080/14656566.2020.1724284                                                DOI: 10.18632/oncotarget.16723                                                       DOI: 10.1080/14656566.2020.1724284

2. In some cases, the manuscript has writing errors that should be checked completely.

Author Response

REVIEWER1:

The manuscript is acceptable after minor revision.

  1. In the introduction, there is no explanation about combination therapy and its benefits, so it is suggested to use the following references:     DOI: 10.22038/ijbms.2016.6649                                                          DOI: 10.1080/14656566.2020.1724284                                                DOI: 10.18632/oncotarget.16723                                                       DOI: 10.1080/14656566.2020.1724284

Authors: we thank the reviewer for the important suggestion, the proposed references have been added in the introduction (citation number 8,9 and 10)

  1. In some cases, the manuscript has writing errors that should be checked completely.

Authors: we are grateful to the reviewer for his careful reading of the manuscript, the entire paper has now been reviewed and writing errors identified have been corrected

Reviewer 2 Report

In the present paper the authors investigated the effects of inhibition of several bromodomain proteins in paclitaxel-resistant ATC cell lines and identified MYCN as a major downstream player in the biological effects of bromodomain proteins inhibition

Major:

1.       Please shortened the introduction section, several sections are abundant and not necessary. As examples lines 46-54; lines 59-68

2.       Please describe in the introduction section the literature relative to bromodomain proteins and thyroid cancer (e.g., 10.3892/ijmm.2018.3642; 10.1158/1078-0432.CCR-16-0914; 10.1016/j.bbrc.2018.03.184)

3.       Bromodomain proteins should be shown in cells resistant or not also at protein levels, antibodies are commercially available

4.       The data on cell viability can be completed with further cell assays describing the kind of death (AnxV/PI stining) and the effects on cell proliferation (BrdU incorporation, cell cycle analysis) at least at the active concentrations

5.       Please provide not only the quantification of wound closure but also images of the assay

6.       Both WNT and cadherin pathways are included in up-regulated and down-regulated group of the RNAseq analysis. Please explain the data providing for example the specific genes up or down modulated

Minor:

1.       Which is the concentration of PTX active on wt 8505c and SW1736 cells?

2.       Do you have evidence of the same mechanisms in cells with a different genetic background? 8505c and SW1736 are both mutated in BRAF

3.       Please discuss why GSK2801 alone increases the wound closure rate

4.       Please discuss differences in 8505c and SW1736 responses to GSK2801 in clonogenic and soft agar assays

5.       Please discuss the higher effects of MYCN inhibition compared to bromodomain protein inhibition

Author Response

REVIEWER2:

In the present paper the authors investigated the effects of inhibition of several bromodomain proteins in paclitaxel-resistant ATC cell lines and identified MYCN as a major downstream player in the biological effects of bromodomain proteins inhibition

Major:

  1. Please shortened the introduction section, several sections are abundant and not necessary. As examples lines 46-54; lines 59-68

Authors: We have shortened the introduction section as requested.

  1. Please describe in the introduction section the literature relative to bromodomain proteins and thyroid cancer (e.g., 10.3892/ijmm.2018.3642; 10.1158/1078-0432.CCR-16-0914; 10.1016/j.bbrc.2018.03.184)

Authors: we briefly described the existence of studies associating brd protein and thyroid cancer. the references suggested by the reviewer have been added (numbers 14,15 and 16)

  1. Bromodomain proteins should be shown in cells resistant or not also at protein levels, antibodies are commercially available

Authors: We performed protein level analyses, by western blot, of the expression of BAZ2A, BAZ2B and BRD9. The results are now shown in Figure 1B and 1C.

  1. The data on cell viability can be completed with further cell assays describing the kind of death (AnxV/PI stining) and the effects on cell proliferation (BrdU incorporation, cell cycle analysis) at least at the active concentrations

Authors: Unfortunately, we are currently unable to conduct further analysis about cell cycle or BrdU incorporation. Hopefully, however, analysis of PARP cleavage levels by western blot (figure 2 D,E) may be sufficient as a marker of cell death type

  1. Please provide not only the quantification of wound closure but also images of the assay

Authors: As requested, the wound healing assay images are now available as Figure 4E

  1. Both WNT and cadherin pathways are included in up-regulated and down-regulated group of the RNAseq analysis. Please explain the data providing for example the specific genes up or down modulated

Authors: the presence of the same two pathways among both the most upregulated and the most downregulated pathways is explained by the up or down regulation of different genes, but belonging to the same pathway. Hoping to make this finding clearer, we have added Table 1 that details which genes in each pathway are more or less expressed after treatment with GSK2801.

Minor:

  1. Which is the concentration of PTX active on wt 8505c and SW1736 cells?

Authors: The concentrations of PTX active on wt 8505C and SW1736 are 4.3 and 3.2 nM respectively. We cited our previous work in which EC50s are established on wt cell lines at the beginning of the results section (citation #3)

  1. Do you have evidence of the same mechanisms in cells with a different genetic background? 8505c and SW1736 are both mutated in BRAF

Authors: To the best of our knowledge, no other paclitaxel-resistant cell models are currently available whose resistance is even partially subverted by administration of GSK2801. However, Yellapu and Bevill demonstrated the efficacy of GSK2801 administered synergistically with other treatments in cells not having mutated BRAF such as HCC-1806, SUM-159, and MDA-MB-468. Both authors' studies are cited in the final part of the introduction (citation n 24, 26)

  1. Please discuss why GSK2801 alone increases the wound closure rate

Authors: one possible explanation for what has been observed is that in this experiment the cells used as controls (PTX) were treated with the dose of paclitaxel necessary to maintain their resistance, whereas the cells treated with GSK2801 alone had no paclitaxel in their culture medium. Although paclitaxel alone has no major effect on these cells because of their resistance, it is possible that it slightly inhibits cell migration under these conditions. If this were the case, cells treated with GSK2801 alone would migrate more not because of it, but because of paclitaxel deprivation. However, the detailed study of the mechanism underlying the change in the migratory capacity of resistant cells is beyond the scope of this study. In addition, in this manuscript, there was no focus on the use of GSK2801 alone as not effective, but only in synergistic use with paclitaxel.

Please discuss differences in 8505c and SW1736 responses to GSK2801 in clonogenic and soft agar assays

Authors: As the reviewer can see from Figure 5B, the two cell lines SW1736-PTX and 8505C-PTX are very different in terms of basal gene expression. We believe that such heterogenicity may explain the difference in response to single treatment with GSK2801. However, we would like to emphasize that despite this diversity at baseline, the use of synergistic treatment, which is the focus of our study, results in almost the same effects, which in our opinion strengthens the data presented.

  1. Please discuss the higher effects of MYCN inhibition compared to bromodomain protein inhibition

Authors: It is complex to establish an exact comparison between the effects of VPC-70619 and GSK2810, because while the dose at which they were tested suggests a greater efficacy of the MYCN inhibitor (2uM vs 20uM), in co-treatment, paclitaxel was used at 500nM with VPC-70619 versus 200nM with GSK2801. Our hypothesis is that BRD9 acts as a modulator of the expression of numerous genes; its inhibition therefore results in the increased expression of some and the reduction of others, including MYCN. It is likely that inhibiting GSK2801, therefore, results in more attenuated effects since the consequent reduction of MYCN is only one of the consequences. Conversely, MYCN inhibition is direct and has only one target, thus showing itself to be more effective. We have argued this point in the discussion.

Reviewer 3 Report

well done

Author Response

Tank you for your time

Reviewer 4 Report

In the paper “GSK2801 reverses paclitaxel resistance in anaplastic thyroid cancer cell lines through MYCN downregulation” the authors describe the effects of inhibition of specific Bromodomain proteins (BAZ2A, BAZ2B and BRD9) via GSK2801 treatments in restoring taxanes sensitivity in Anaplastic Thyroid PTX resistant cells. By evaluating GSK2801 effects, the authors also proposed MYCN downregulation as an important step to restore taxanes sensitivity in anaplastic thyroid PTX resistant cells.

General comments: the author should include Paclitaxel sensitive cells in each experiment of the paper as control to further prove and sustain their results. Moreover, can the author include an experiment to prove that they are really inhibiting GSK2801 targets? (e.g. protein localization, the binding to chromatin).

Major concerns:

Point 1- In Figure 1, only BRD9 mRNA expression is upregulated in Paclitaxel resistant cells while the previous study (Allegri et al., 2021) “ ..had shown expression..” (lines 197-198) of GSK2801 targets in paclitaxcel resistant cells.

The authors should include a western blotting of GSK2801 targets (BRD9, BAZ2a and BAZ2B) in Paclitaxel sensitive and resistant cells to confirm the presence of higher levels of these proteins after development of drug resistance and the eventual differences in protein expression between the resistant used cell lines. This experiment is necessary to better correlate LEVELS of Bromodomain proteins to the different behaviors of 85005C-PTX and SW1736-PTX after drug exposure (later explained in Fig 2, 3).

Point 2- In Figure 2, the authors should include a growth curve of paclitaxel sensitive and resistant cells treated with a concentration of 20 uM of GSK2801 and a higher concentration (at least till 40 uM) This experiment is necessary because in Fig 2B they tested viability of cells treated with 20 or 40 uM GSK2801 in combination with paclitaxel.

Can the author explain why they choose the drug combination PTX200 nM/ GSK 20 uM and not PTX 100nM/GSK 40uM?

The authors should evaluate whether the toxicity of paclitaxel and GSK2801 in combination is due to additive or synergistic effects. To this aim, they could calculate the Ic50 values of the two drugs alone and the combination index of the combined treatment (Ting-Chao Chou, Cancer Res. 2010 doi: 10.1158/0008-5472).

Moreover, did the author evaluate whether depletion of BRD9 in Paclitaxel resistant cells mimic (at least in part) GSK2801 effects in combination with PTX treatment?

Point 3- In Figure 3, 85005C-PTX cells already showed significant reduced clonogenic capacity after GSK2801 treatment (20 uM?). How Do the authors explain such result? Can they correlate BRD9 protein expression levels and GSK2801 sensitivity? (see point 1)

Clonogenic capacity of paclitaxel untreated-PTX resistant cells and Paclitaxel untreated and treated PTX SENSITIVE cells should be included in the experiment of Figure 3.

Can the author also evaluate the dimension of colonies? Are the colonies smaller (e.g. formed by few cells) when resistant cells are treated with the combination of GSK/PTX compared to GSK alone?

Point4- The second part of the paper results in some points difficult to follow and sometimes experiments are not clearly presented (see lines 341-345).  Can the authors confirm (by RT-PCR) the reduced expression of MYCN gene after VPC70619 treatment? Can the authors confirm that some of MYCN targets are downregulated after VPC70619 treatment?

Can the authors evaluate expression levels of MYCN in Paclitaxel resistant and sensitive cells?

What are the effects of VPC70619 treatment on Paclitaxel sensitive cells? If GSK2801 “reverses paclitaxel resistance in anaplastic thyroid cancer cell lines through MYCN downregulation” why GSK2801 treatment in Paclitaxel resistance cells did not show any effects on cell viability? (see point 2).

Minor concerns:

In figure 4A the anchorage independent capacity of the cell lines used in the paper is opposite when treated with only GSK2801. The author should comment this result. Is there any information about the aggressiveness of such cell lines? Are these cell lines activating EMT differently?

In figure 4B can the author show a representative picture of the wound healing assay? Why the treatment alone of GSK2801 is increasing the wound healing capacity? The author should comment this result.  

In Figure 7 the number of colonies counted in paclitaxel treated 85005C-PTX and SW1736-PTX cells is significant higher compared to the number of colonies counted in same experimental condition shown in figure 3. I strongly suggest to repeat this experiment.

Author Response

REVIEWER 4:

In the paper “GSK2801 reverses paclitaxel resistance in anaplastic thyroid cancer cell lines through MYCN downregulation” the authors describe the effects of inhibition of specific Bromodomain proteins (BAZ2A, BAZ2B and BRD9) via GSK2801 treatments in restoring taxanes sensitivity in Anaplastic Thyroid PTX resistant cells. By evaluating GSK2801 effects, the authors also proposed MYCN downregulation as an important step to restore taxanes sensitivity in anaplastic thyroid PTX resistant cells.

General comments:

  • the author should include Paclitaxel sensitive cells in each experiment of the paper as control to further prove and sustain their results.
  • -Moreover, can the author include an experiment to prove that they are really inhibiting GSK2801 targets? (e.g. protein localization, the binding to chromatin).

Major concerns:

-Point 1- In Figure 1, only BRD9 mRNA expression is upregulated in Paclitaxel resistant cells while the previous study (Allegri et al., 2021) “ ..had shown expression..” (lines 197-198) of GSK2801 targets in paclitaxcel resistant cells.

The authors should include a western blotting of GSK2801 targets (BRD9, BAZ2a and BAZ2B) in Paclitaxel sensitive and resistant cells to confirm the presence of higher levels of these proteins after development of drug resistance and the eventual differences in protein expression between the resistant used cell lines. This experiment is necessary to better correlate LEVELS of Bromodomain proteins to the different behaviors of 85005C-PTX and SW1736-PTX after drug exposure (later explained in Fig 2, 3).

Authors: Western blot analysis of protein levels of BAZ2A, BAZ2B and BRD9 in non-paclitaxel resistant and paclitaxel resistant cell lines has been added to Figure 1 (1B and 1C)

Point 2- In Figure 2, the authors should include a growth curve of paclitaxel sensitive and resistant cells treated with a concentration of 20 uM of GSK2801 and a higher concentration (at least till 40 uM) This experiment is necessary because in Fig 2B they tested viability of cells treated with 20 or 40 uM GSK2801 in combination with paclitaxel

Authors: cell viability assay after treatment with GSK2801 in SW1736 and 8505C cells not resistant to paclitaxel was added in Figure 2 (2A). For both sensitive and resistant cells, the effects are also shown for doses 20 and 40 uM (2A and 2B)

Can the author explain why they choose the drug combination PTX200 nM/ GSK 20 uM and not PTX 100nM/GSK 40uM?

Authors: Undoubtedly the reviewer's observation is correct, however since that SW1736-PTX and 8505C-PTX cells show an paclitaxel-ED50 greater than 5-10 µM, we preferred to focus on the lower dose with greater effect of GSK2801.

The authors should evaluate whether the toxicity of paclitaxel and GSK2801 in combination is due to additive or synergistic effects. To this aim, they could calculate the Ic50 values of the two drugs alone and the combination index of the combined treatment (Ting-Chao Chou, Cancer Res. 2010 doi: 10.1158/0008-5472).

Authors: We had calculated the combination index by using CompuSyn software based on the paper suggested by the reviewer (Ting-Chao Chou, Cancer Res. 2010 doi: 10.1158/0008-5472). Since SW1736-PTX and 8505C-PTX cells are by definition resistant to paclitaxel, at the doses studied, the program is unable to calculate the ED50/IC50 of paclitaxel (none of these doses have the slightest effect). On the other hand, increasing paclitaxel doses by pushing up to those close to the ED50/IC50 of resistant cells, no gain of effect is observed in combinations with GSK, as already with paclitaxel concentrations of 200nm the gsk 20 and 40 µm results in an almost total effect. The follow image shows output of the program used, if the reviewer agrees, we would prefer not to publish this result because in our opinion does not add much to the data about the synergistic action of GSK2801 and paclitaxel.

Moreover, did the author evaluate whether depletion of BRD9 in Paclitaxel resistant cells mimic (at least in part) GSK2801 effects in combination with PTX treatment?

Authors: Unfortunately, we are currently unable to perform a silencing experiment of BRD9. However, for example, the use of siRNAs directed against BRD9 or the creation of a stable deletion mutant for BRD9 would result in non-expression of the protein. This could lead to indirect side effects, either intrinsic to the method used or due to BRD9 depletion. The use of a specific inhibitor capable of displacing its binding to its targets, on the other hand, represents in our opinion a more fine-grained approach, which allows to focus more on the main action of BRD9 analyzed in this study, namely that of interaction with its targets to regulate its expression. Certainly, in future studies, the complete or almost complete absence of BRD9 will also be tested.

Point 3- In Figure 3, 85005C-PTX cells already showed significant reduced clonogenic capacity after GSK2801 treatment (20 uM?). How Do the authors explain such result? Can they correlate BRD9 protein expression levels and GSK2801 sensitivity? (see point 1)

Authors: The different response of the two cell lines to treatment with GSK2801 alone could be due to a multiplicity of factors, first among them the difference in transcriptional pattern observable, for example, by transcriptomic analysis. One hypothesis, as brilliantly pointed out by the reviewer, could be the lower basal expression of BRD9 in 8505C-PTX cells as can be seen from Figure 1B. A lower amount of BRD9 could mean a better efficacy of GSK2801 at the same dose used, however, this phenomenon is only observed in terms of the plate colony formation assay. In the soft agar assay the situation is mirrored with the SW1736-PTX while in migration nothing similar is observed. All this, combined with the fact that there is no effect in terms of cell viability/proliferation makes us think of a phenomenon that is not relevant in the overall balance of the effects of GSK2801 alone in this model

Clonogenic capacity of paclitaxel untreated-PTX resistant cells and Paclitaxel untreated and treated PTX SENSITIVE cells should be included in the experiment of Figure 3.

Can the author also evaluate the dimension of colonies? Are the colonies smaller (e.g. formed by few cells) when resistant cells are treated with the combination of GSK/PTX compared to GSK alone?

Authors: We take this interesting observation by the reviewer to respond to an earlier criticism that, according to the reviewer, would affect every experiment in this study. In this paper, SW1736 and 8505C (paclitaxel-sensitive) cells were never considered (except for the cell viability assay after treatment with GSK2801 alone in Figure 2). This responds to a specific choice of the authors to limit the approach to deal with the problem concerning ATC cells already resistant to paclitaxel. Moreover, there would exist an underlying problem that would be difficult to overcome, since at the concentrations of paclitaxel used in this study (e.g., 100-200nM) paclitaxel-sensitive cells would not resist. In fact, the ED50 of paclitaxel-sensitive cells is around 5 nM, which would make any experimental comparison impossible (the different cell lines would have to be treated with doses two orders of magnitude different, this would result in an uninformative data).

In response, however, to the second observation, colony size had been measured, but since there was no significant variation, this data was not included.

Point4- The second part of the paper results in some points difficult to follow and sometimes experiments are not clearly presented (see lines 341-345).  Can the authors confirm (by RT-PCR) the reduced expression of MYCN gene after VPC70619 treatment? Can the authors confirm that some of MYCN targets are downregulated after VPC70619 treatment?

Authors: As requested by the Reviewer, in figure 8 we added expression evaluation of some  MYCn target genes after VPC-706091 treatment.

Can the authors evaluate expression levels of MYCN in Paclitaxel resistant and sensitive cells?

What are the effects of VPC70619 treatment on Paclitaxel sensitive cells? If GSK2801 “reverses paclitaxel resistance in anaplastic thyroid cancer cell lines through MYCN downregulation” why GSK2801 treatment in Paclitaxel resistance cells did not show any effects on cell viability? (see point 2).

Autohors: the point made by the reviewer is certainly pertinent. As previously explained, the purpose of this study is the possibility of making paclitaxel-resistant anaplastic thyroid cancer cells sensitive again, therefore, to lay the groundwork for addressing the serious, and currently unresolvable, problem of resistance to therapy that these tumors develop in vivo. From previously performed RNA-seq, we find a gain in MYCN expression in resistant cells compared with susceptible cells; it goes from near-zero to detectable values. Given that, in this study it is shown that MYCN is a target of BRD9 and that inhibition of BRD9 binding activity also inhibits MYCN binding and down regulates MYCN expression. This finding, together with what the reviewer rightly observed (GSK2801 down regulates MYCN, but has no effect on its own), allows us to formulate the hypothesis that MYCN only intervenes in the acquisition/maintenance of paclitaxel resistance. Consequently, MYCN does not appear to be influential in paclitaxel-sensitive tumor cells or in resistant cells when paclitaxel is not administered.

Minor concerns:

In figure 4A the anchorage independent capacity of the cell lines used in the paper is opposite when treated with only GSK2801. The author should comment this result. Is there any information about the aggressiveness of such cell lines? Are these cell lines activating EMT differently?

Authors: Similarly to what argued for the effects of GSK2801 alone on plate colony formation in 8505C-PTX cells, we believe that further studies that address in more detail the mechanisms of anchorage-dependent clonogenicity are required. What we can hypothesize here is that there is strong heterogeneity in the gene expression background between the two cell lines, which may explain different behaviors. We consider the effect of GSK2801 alone not relevant in our study, since its message is  about the use of GSK2801 in synergy with paclitaxel, a combination treatment that, unlike single treatment, gives consistent and clear-cut results in both types of colony formation assay. Regarding EMT status, we have previously evaluated it and, based on expression of some EMT markers  cell lines here utilized show mesenchymal-like phenotype (low expression levels of CDH1, high expression levels of CDH2, vimentin, TWIST, ZEB etc.).

In figure 4B can the author show a representative picture of the wound healing assay? Why the treatment alone of GSK2801 is increasing the wound healing capacity? The author should comment this result.

Authors: The images have now been added. Regarding the effects of GSK2801 alone this could be due to the fact that compared to the paclitaxel control (PTX), paclitaxel was removed. Although these cells are resistant to paclitaxel in every respect as already published and reconfirmed in this study, it is possible that high-dose paclitaxel constantly added to the culture medium in which the cells are grown induces a slight reduction in motility as measured by this type of assay. the effects, if so, would then be due to a deprivation of paclitaxel rather than direct effects of GSK2801 alone. However, this is significant in only one of the two cell lines, and again confirms how the use of GSK2801 should always be considered in synergy with paclitaxel.

In Figure 7 the number of colonies counted in paclitaxel treated 85005C-PTX and SW1736-PTX cells is significant higher compared to the number of colonies counted in same experimental condition shown in figure 3. I strongly suggest to repeat this experiment.

Authors: experiment was repeated, the number of colonies of the controls in the two figures (Figure 3 and Figure 9-ex7) are now comparable being in both cases between 450 and 650.

Round 2

Reviewer 2 Report

The authors have satisfactorily addressed most of my concerns

Reviewer 4 Report

In this second version of the paper “GSK2801 reverses paclitaxel resistance in anaplastic thyroid cancer cell lines through MYCN downregulation” the manuscript has been improved and additional experiments have been included to further sustain the results. In particular, the authors not only included many controls (western blotting of Bromodomain proteins in paclitaxel resistant and sentitive cells, representative pictures of colony assay and migration assay, evaluation of myc target genes) but also clearly demonstrated the specific ability of GSK2801 to impair BRD9 binding to several targets, among all MYC gene.